# Enhanced Impact of Land Reclamation on the Tide in the Guangxi Beibu Gulf

Jingfang Lu [1], Yibo Zhang [1], Ruichen Cao [1], Xianqing Lv [1], Minjie Xu [2,*], Guandong Gao [3,4,5] and Qiang Liu [6]

[1] Frontier Science Center for Deep Ocean Multispheres and Earth System (FDOMES) and Physical Oceanography Laboratory, Ocean University of China, Qingdao 266100, China; lujingfang@stu.ouc.edu.cn (J.L.); zhangyibo9300@stu.ouc.edu.cn (Y.Z.); caoruichen@stu.ouc.edu.cn (R.C.); xqinglv@ouc.edu.cn (X.L.)

[2] Ocean School, Yantai University, Yantai 264005, China

[3] CAS Key Laboratory of Ocean Circulation and Waves, Institute of Oceanology Chinese Academy of Sciences, Qingdao 266071, China; guandonggao@qdio.ac.cn

[4] Function Laboratory for Ocean Dynamics and Climate, Qingdao National Laboratory for Marine Science Technology, Qingdao 266237, China

[5] Center for Mega-Science, Chinese Academy of Sciences, Qingdao 266071, China

[6] College of Engineering, Ocean University of China, Qingdao 266100, China; liuqiang@ouc.edu.cn

\* Correspondence: minjiexu@ytu.edu.cn

**Abstract:** Based on the method for identifying the boundary of movable water bodies (MWBB), the spatial distribution of reclamation projects in the Guangxi Beibu Gulf were identified over the past 40 years and the impact of these engineering facilities on hydrodynamics was also evaluated. The results showed that 163.8 km$^2$ of natural sea areas in the Guangxi Beibu Gulf were occupied through reclamation in the last 40 years. The effects of land reclamation on tidal amplitude were more pronounced in the second period (2001–2018) than in the first period (1987–2001), particularly in the tidal channels of Qinzhou Bay and Fangcheng Bay, where the amplitude difference ranged from 8 to 15 cm, representing a 40–55% increase. The reduction in the sea area because of land reclamation has changed the hydrodynamics in the Guangxi Beibu Gulf, including reducing the tidal volume, altering the amplitude variations, and increasing the seaward residual currents, all of which could cause significant problems for the coastal environment.

**Keywords:** Guangxi Beibu Gulf; movable water bodies boundary; reclamation projects; spatio-temporal characteristics; hydrodynamics; drivers





## 1. Introduction

The coastal zone, as a transitional area where land and sea intersect, due to its superior geographic location, rich natural endowment, multiple resources, and suitable climatic conditions, has attracted a large number of people and production factors to gather around it [1,2]. Coastal reclamation is usually the first choice in order to gain more land for residence, commercial development, tourism and leisure, food and energy aquaculture, as well as oil and gas exploitation, even though it changes the natural properties of the ocean through ocean filling with sand, soil, stone, a cement mixture, etc., or dam blockage [3,4]. At present, globally, the marine construction boom driven by coastal city expansions, population growth, and economic development, including construction in the marine environment for various purposes (e.g., ports, cross-sea bridges, aquaculture ponds), also causes changes in coastlines and hydrodynamics [5,6]. Studies have shown that reclamation-induced coastline change can weaken the tidal current [7,8], reduce the tidal prism [9,10], and result in tidal amplitude and phase lag variations [11–13]. Even though coastal reclamation benefits society greatly, it also creates significant challenges for coordinating economic development and the marine environment.

Hydrodynamics (sea level, wave conditions, storm surge, coastal currents, and river flow) is a primary driver of coastal change [14,15]; meanwhile, shoreline changes affect the simulation of physical parameters, such as the tidal range, current, and waves. Lin et al. [16] proved that a change in the shorelines made the residual current in the west channel increase by up to 0.10 (0.05) m/s in the surface (near bottom) layer in the Pearl River Estuary, China. In Tampa Bay, channel deepening and widening increase the nontidal circulation, resulting in an increase in salinity [17]. In addition, the study confirmed that harbors, seawalls, industrial complexes, and urban districts constructed on the reclaimed land will permanently change the geomorphology of the coast line and the physical processes in the coastal system, resulting in more impacts on the coastal hydrodynamics and sediment processes [18]. Reclamation activity-induced shoreline changes in the coastal area of Jupo, Yeongsan Estuary, South Korea, resulted in the reduction of tidal backwater, and the expansion of the tidal range [19]. Wishaa et al. [20] indicated that the current velocity decreases proportionally in line with the reclamation development in Benoa Bay water, Bali, Indonesia. Zeng et al. [21] verified that the tidal prism near Xiangshan Port has been significantly reduced due to land reclamation. A few studies have been conducted on the hydrodynamic changes due to bridge construction. The Jiaozhou Bay Bridge in Qingdao was built in 2011, with a length of 26.75 km. The study results indicated that the tidal current changed obviously, because the accumulated blockage effect of the whole bridge induced a tidal change at the bay entrance [22,23].

The Guangxi Beibu Gulf Economic Zone was developed relatively late and with a lower degree of marine resource development than other coastal areas in China, due to its unique geographic location. However, after years of development and construction, the Guangxi Beibu Gulf Economic Zone has been transformed from a remote western border area into a multi-regional cooperation center and a leading region for open development, and has been included in the national development strategy [24]. For example, at the end of 2019, the Guangxi Beibu Gulf Economic Zone had a total population of 24.56 million, with an annual GDP of CNY 103.42 billion, accounting for 48.4% of Guangxi (Guangxi Zhuang Autonomous Region Statistical Yearbook, 2020). Along with the rapid development of the Beibu Gulf Economic Zone, the reclamation activities have already caused the hydrodynamic environment of the coastal water to undergo a series of changes. For example, Wang et al. [25] investigated the effects of coastline change on the dynamics and water exchange in the Maowei Sea (part of the Beibu Gulf) from 2008 to 2018, and found that the tidal prism decreased and the water exchange half-life increased after reclamation. Though their work is significant, studies on the long-term temporal variability in coastlines and tides remain insufficient. Lu et al. [26] developed a Bohai tidal evaluation model based on the optimum position of remotely observed coastlines, and the model results were characterized by decades of long-term temporal features. On the basis of the previous work, this paper adopted a fine-grid numerical model in order to assess the impact of land reclamation on narrow structures, such as the tidal channels between Qinzhou Bay and that Maowei Sea that are not fully resolved in conventional models.

In this study, we combined a recently developed movable water bodies boundary extraction (MWBB) method, with data assimilation of a tidal model that filters individual boundary pixels into a consistent tidal numerical model using an instantaneous boundary from the moment of satellite image acquisition. Further, the structural equation model (SEM) technique systematically explained the reclamation mechanism and analyzed the effects of coastal reclamation on population growth, economic development, and marine industry development. The goals of this study were: (1) to draw the spatio-temporal changes of the MWBB and reclamation in the Guangxi Beibu Gulf; (2) to explore whether regime shifts in tidal dynamics can occur in a semi-enclosed gulf, such as the Beibu Gulf, under reclamation impact; (3) to assess the evolution of the potential effects from reclamation projects; and (4) to explain the driving mechanism of reclamation. The results will provide a decision-making reference for monitoring coastal change across dynamic coastal zones in the Guangxi Beibu Gulf.

## 2. Materials and Methods

### 2.1. Study Area

The Guangxi Beibu Gulf is located in the northwestern part of the South China Sea (Figure 1). The scope of this study is bounded by the latitudes 21°N and 22°N, and by the longitudes 108°E and 122°E. According to our research, from 1995 to 2015, the reclaimed area of Guangxi Beibu Gulf showed continuous growth (Figure 2), with an average of 4.7 km² per year [27]. While these coastal projects have undoubtedly contributed to economic progress, they have also inevitably had potential impacts on the coastal zones, such as alterations to the coastline, land-use conversions, and variations in the marine hydrodynamics.

The Beibu Gulf was mainly affected by three tidal waves: $O_1$, $K_1$, and $M_2$. The largest amplitudes of the $O_1$, $K_1$, and $M_2$ tides can reach 100 cm, 90 cm, and 60 cm, respectively [28]. Hence, we only focus on these three tides (the $M_2$, $K_1$, and $O_1$ tides) in this study. The amplitude of the tides in the Beibu Gulf varies considerably, ranging between 0.5 to 8 m. This variability in amplitude is mainly influenced by several geographical and meteorological factors, such as seabed topography, atmospheric pressure, and wind patterns [29]. In addition, the complex coastline also affects the tide's amplitude, as it causes the tide to change direction and flow in various directions, resulting in higher localized heights. The Beibu Gulf's topographic features, which include shallow waters, narrow channels, and complex coastlines, play a significant role in amplifying the tide's amplitude.

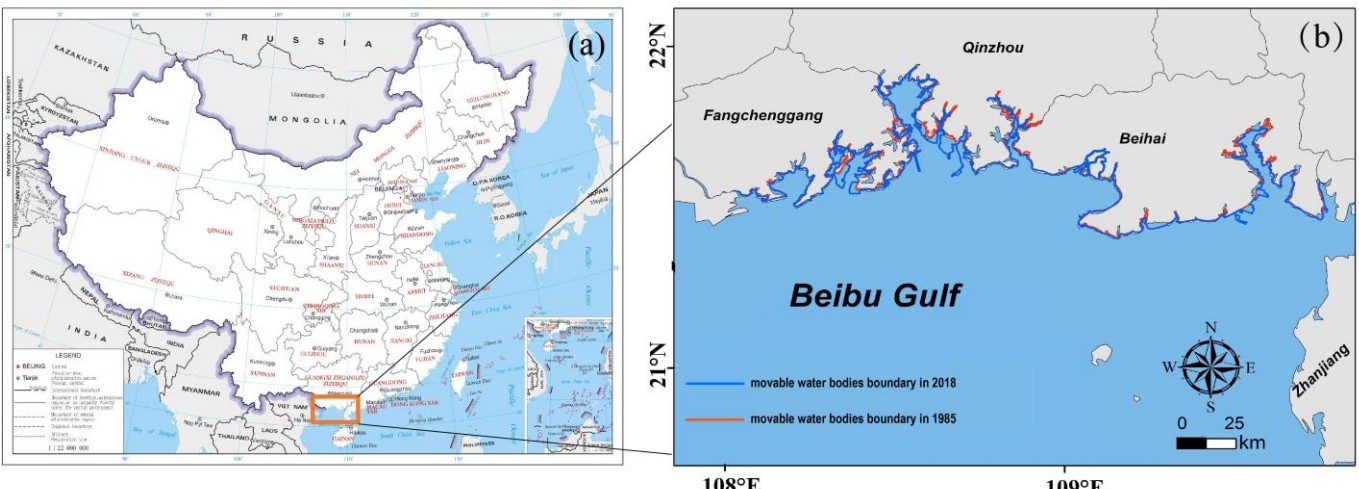

**Figure 1.** Location of the studied area. (**a**) The map of China in the upper left corner downloaded from http://bzdt.ch.mnr.gov.cn/ (accessed on 3 June 2023). (**b**) The coastline underwent significant changes from 1987 to 2018, as a result of coastal reclamation activities.

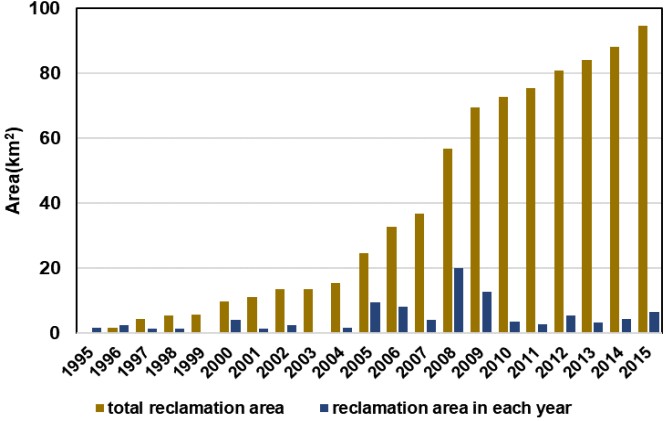

**Figure 2.** The changes in reclamation area from 1995 to 2015, based on Lu et al. [27].

### 2.2. Overall Workflow

With the rapid population growth and the increase in economic development in the Guangxi Beibu Gulf, converting natural coastlines into artificial coastlines for port development, aquaculture ponds, and industrial purposes was inevitable. Moreover, the length and shape of coastlines are changed greatly by coastal reclamation. Long-term human development activities can effectively alter coastal topography, subsequently reducing or increasing the tidal range, energy, and current velocity. Accordingly, a conceptual framework to study the impact of reclamation on the tidal environment is put forward (Figure 3).

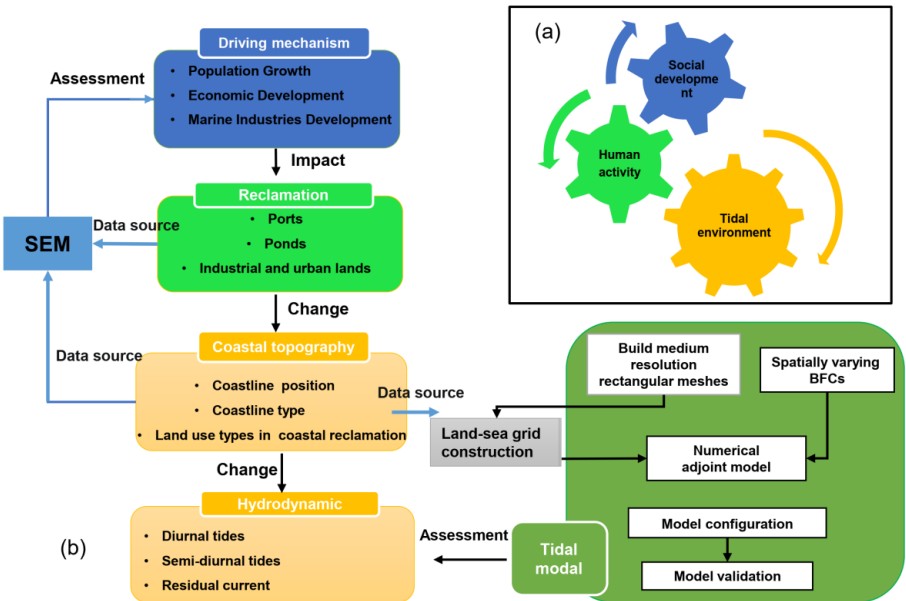

**Figure 3.** The overall workflow of the method in this study. (**a**) Driving effect. (**b**) Technical route.

### 2.3. Data Source

The data used in this study are mainly from the following four resources: Landsat satellite data (Table 1) with a spatial resolution of 30 m are widely used for shoreline identification and assessment and have been proven to be a reliable method [30,31]; the Radar Altimeter Database System (http://rads.tudelft.nl, accessed on 6 July 2023) provides access to the satellite altimeter data used in our investigation, including the tidal harmonic constants derived for 25 years from TOPEX/Poseidon (T/P), which was used in a two-dimensional tidal model of the Guangxi Beibu Gulf; the ETOPO1 ocean bathymetry data, which had a spatial resolution of $1/60° \times 1/60°$ and was released in October 2020 through https://data.nodc.noaa.gov (accessed on 13 July 2023); the bathymetry data for the inner bay, which was obtained from marine charts published by the China Maritime Safety Administration; and the statistical data from the Guangxi Statistical Yearbooks (1995–2015) [32] and the China Marine Statistical Yearbook (1995–2015) [33].

**Table 1.** The remote sensing image data on the Guangxi Beibu Gulf.

| Region | Year | Image Sensor | Path/Row |
|--------|------|--------------|----------|
| Beibu Gulf | 1987, 1994, 1995–2015, 2018 | Landsat5 TM, Landsat7 ETM, Landsat8 OLI | 125/045, 124/045, 124/046 |

### 2.4. Methods

2.4.1. MWBB Monitoring and Extraction

The coastline obtained through the remote sensing automatic classification method essentially represents the instantaneous water's edge at the time when the satellite passes

over. Its location varies greatly due to factors such as tides and coastal topography [34]. Therefore, in order to scientifically reflect the dynamic changes in the coastline, the manual visual interpretation method was used to determine the position of the coastline. The "streaming" tool was used to outline the coastline, leaving a coordinate point every 0.1 s, with a distance error between points less than 5 m. A total of 224,305 coordinate points were generated and stored in longitude and latitude format.

This paper employed the delineation method by Lu et al. [35] for the movable water bodies boundary (MWBB), which was defined as the solid boundary blocking the movement of the sea and destroying its connectivity at the intersection of land and sea. Considering the actual remote sensing image of the Guangxi Beibu Gulf, its MWBB was divided into seven categories: the bedrock boundary, the mangrove boundary, the estuary boundary, the industrial and urban boundary, the pond boundary, the protective dam boundary (including seawalls and groynes), the port boundary, and permeable bridges (Figure 4). The defining methods for MWBBs with different utilization types was illustrated in our previous study [35]. In this study, we further clarified the definition of boundaries and extraction methods for mangroves, aquaculture ponds, and permeable bridges, as follows:

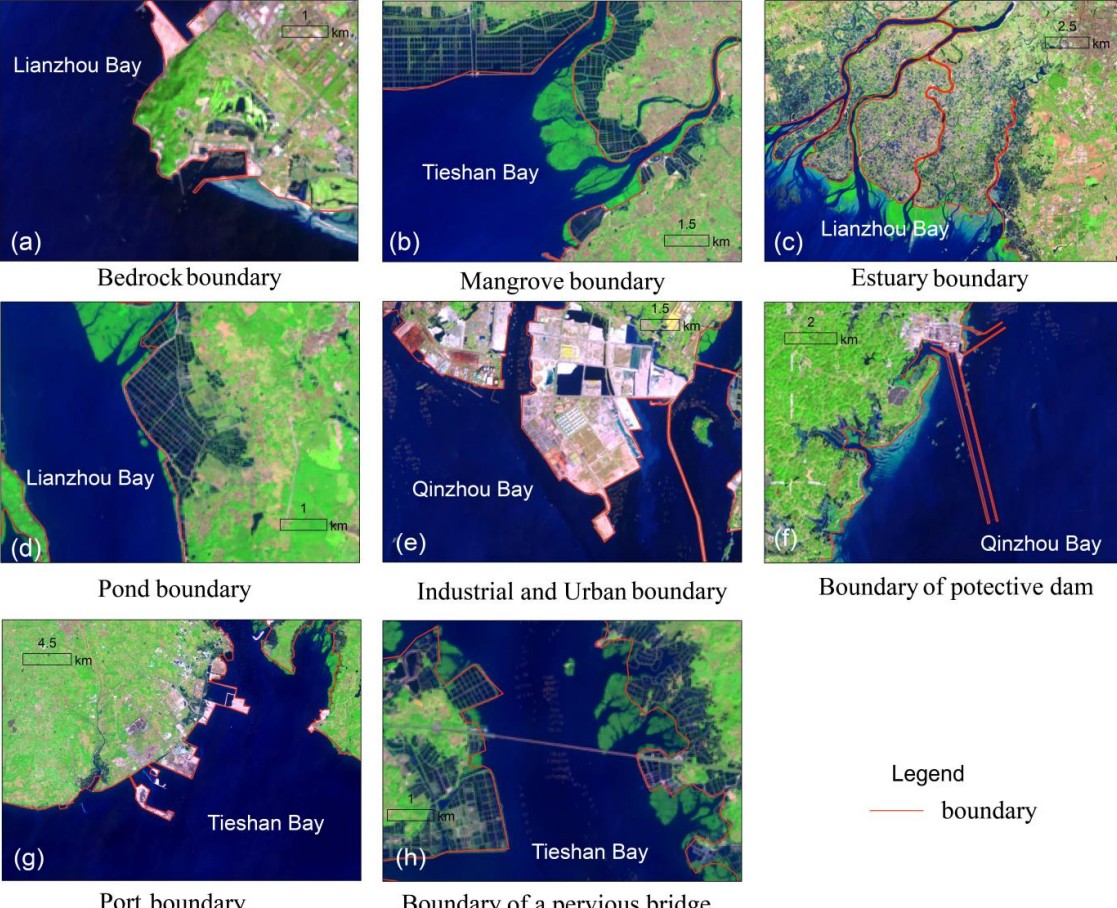

**Figure 4.** MWBB interpretation. (**a**) Guantou Mountain, part of Lianzhou Bay's bedrock boundary; (**b**) a mangrove, located in Tieshan Bay; (**c**) Nanliu River Estuary, located in Beihai City (**d**) a mariculture pond, located in Lianzhou Bay; (**e**) Qinzhou Bonded Port, South China's second largest port after Guangzhou Port; (**f**) two breakwaters nearly 7 km long, connected to the land and designed to protect the land, located in Qinzhou Bay; (**g**) three port areas built in Tieshan Bay; (**h**) boundary of the Tieshan Bay Bridge, located in Beihai City.

(1) For mangrove distribution areas, they appear as irregular patches, green in color, clearly distinguishable from the deep brown color of the surrounding land bodies.

The contour lines of the mangroves are chosen as the boundary lines since they are typical of the intertidal zone.

(2) For aquaculture ponds and reservoirs, due to the dam being higher than the high tide level, the seawater cannot pass over the dam during high tide. When the floodgate is opened, the rate of water exchange is slow, and when the floodgate is closed, the water body is isolated from the seawater. Overall, the water body inside the dam is considered a non-natural dynamic water area, therefore the boundary line is determined by their seaward side.

(3) For permanent coastal structures, such as ports, seawalls, and groynes, the boundary line is determined by the seaward side; whereas, for permeable bridges, although the bridge piers may affect the seawater exchange, the overall water exchange remains smooth, and the boundary line is determined by the landward side.

### 2.4.2. Tidal Model

Based on the tidal simulation method by Lv and Zhang [36], the $M_2$ tide in the Guangxi Beibu Gulf was simulated through adjoint assimilation, and the tidal harmonic constants at the open boundary were inverted based on the T/P data extracted from the sea area. The numerical scheme of forward equations and adjoint equations is included in the Supplementary Material.

(1) The governing equations for the vertically integrated equations of continuity and momentum are as follows:

$$\frac{\partial \zeta}{\partial t} + \frac{1}{a}\frac{\partial[(h+\zeta)u]}{\partial \lambda} + \frac{1}{a}\frac{\partial[(h+\zeta)v\cos\phi]}{\partial \phi} = 0,$$

$$\frac{\partial u}{\partial t} + \frac{u}{a}\frac{\partial u}{\partial \lambda} + \frac{v}{R}\frac{\partial u}{\partial \phi} - \frac{uv\tan\phi}{R} - fv + \frac{ku\sqrt{u^2+v^2}}{h+\zeta} - A\Delta u + \frac{g}{a}\frac{\partial(\zeta-\bar{\zeta})}{\partial \lambda} = 0,$$

$$\frac{\partial v}{\partial t} + \frac{u}{a}\frac{\partial v}{\partial \lambda} + \frac{v}{R}\frac{\partial v}{\partial \phi} + \frac{u^2\tan\phi}{R} + fu + \frac{kv\sqrt{u^2+v^2}}{h+\zeta} - A\Delta v + \frac{g}{R}\frac{\partial(\zeta-\bar{\zeta})}{\partial \phi} = 0.$$

where $t$ is the time, $\lambda$ and $\phi$ are the east longitude and north latitude, respectively, $\zeta$ is the sea surface elevation above the undisturbed sea level, $u$ and $v$ are the east and north components of the fluid velocity, respectively, $\bar{\zeta}$ is the adjusted height of the equilibrium tides, $R$ is the radius of the earth, $a = R\cos\phi$, $f = 2\Omega\sin\phi$, where $\Omega$ represents the angular speed of the Earth's rotation, $g$ is the acceleration due to gravity, $h$ is the undisturbed water depth, and $h + \zeta$ denotes the total water depth, $k$ is the bottom friction coefficient, $A$ is the coefficient of the horizontal eddy viscosity, $\Delta$ is the Laplace operator, and $\Delta(u,v) = a^{-1}[a^{-1}\partial_\lambda(\partial_\lambda(u,v)) + R^{-1}\partial_\phi(\cos\phi\partial_\phi(u,v))]$.

Model setting: The model domain is 21°06′N to 21°54′N and 108°02′E to 109°47′E, and the horizontal resolution is set to 1/60°. The harmonic constants at the 84 satellite altimeter cross-over points (from Pan et al. [37]) are used as observations to optimize the two coefficients (amplitude and phase lag) with the adjoint method. The closed boundary conditions for our model are zero flow normal to the coast. That is, $\vec{u} \cdot \vec{n} = 0$ for the grid points at the closed boundary, where $\vec{n}$ is the outward unit vector and $\vec{u} = (\mathbf{u},\mathbf{v})$ is the velocity vector.

Along the open boundaries, the water elevation of the tide at the *jth* time step is given as $\zeta^j_{m_l,n_l} = a_{0,l} + [a_l\cos(\omega j\Delta t) + b_l\sin(\omega j\Delta t)]$, where $(m_l, n_l)$ stands for the grid points at the open boundaries, $\omega$ is the frequency of the tidal constituent, and $a_l,b_l$ are the Fourier coefficients.

Model verification: The model fidelity was confirmed by comparing it with the tidal harmonic constants observed from the 84 tide gauges. For $M_2$, $O_1$, and $K_1$, the absolute mean error (MAE) deviations between the model results and the observations are 4.0 cm, 3.4 cm, and 6.0 cm, respectively. For the phase lag, the MAEs are 11.7°, 5.7°, and 7.8°, respectively. The MAEs of the two parameter values meet the requirements [38]. Additionally, for the $R^2$ results for $M_2$ in 1987, 2001, and 2018, the simulated results have a significant correlation with the measurements (Figure 5).

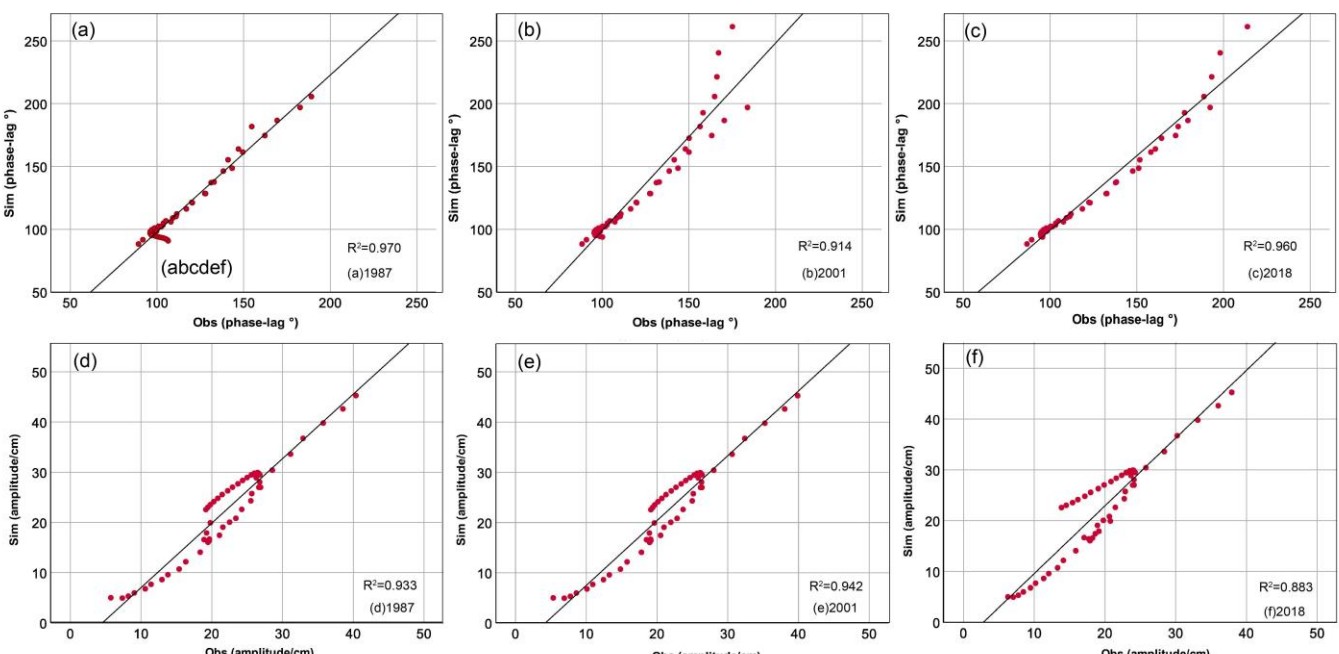

**Figure 5.** Deviations between the simulated results and observations for three years, (**a**–**c**) denote phase lag (°), and (**d**–**f**) denote amplitude (cm).

2.4.3. Statistical Analysis of Data

Following Yu et al. [39], we used structural equation modeling (SEM) to identify the existence of the driving mechanism of coastal land reclamation. In general, SEM is a statistical methodology that enables the investigation of direct and indirect relationships among causal variables through a single model, as well as the testing of a conceptual model detailing the relationship between observed and latent variables [40,41]. The studies showed that the growth in reclaimed land was mainly driven by social factors (in the coastal region), such as the rapid development of the economy and population, and the rapid expansion of marine construction (e.g., aquaculture ponds, ports, and construction land) [4]. As a result, we selected the following indicators to investigate their relationship with the growth in reclaimed land.

Index selection: We employed 8 indexes collected from the Guangxi Statistical Yearbooks (1995–2015), the China Marine Statistical Yearbook (1995–2015), and remote sensing interpretation data. Six of them were placed into the first layer of the three indicators: population growth, economic development, and marine industry development. In the Guangxi Beibu Gulf, the need to expand into the sea is demanded due to the economic development, population growth, and the requirement for agriculture, and other industries [42]. In general, an SEM model was established with the statistical data and reclamation vector data in twenty-one survey periods; refer to Table S1 for the data pertaining to the index.

## 3. Results

### 3.1. Spatio-Temporal Characteristics of MWBB

3.1.1. MWBB's Location Change

From 1987 to 2018, the reclaiming of the Guangxi Beibu Gulf had already resulted in significant spatio-temporal changes to the MWBB (Figure 6). Our findings reveal that 23.1% of the MWBB in the Guangxi Beibu Gulf, mapped in 2018, was seaward movement compared with its position in 1987; the MWBB change was mainly concentrated in Qinzhou Bay, Tieshan Bay, and Fangcheng Bay. We further calculated the changes in the coastline in the study area based on the end-point rate method [30]. Beibu Gulf's shoreline changed at a rate of 12.1 m/a, on average, over the previous 30 years. The coastline of Fangcheng Bay showed a trend of accreting toward the sea and exhibited the fastest average change

(34.3 m/a) in the Beibu Gulf, followed by Qinzhou Bay (27.1 m/a), and Tieshan Bay (20.1 m/a).

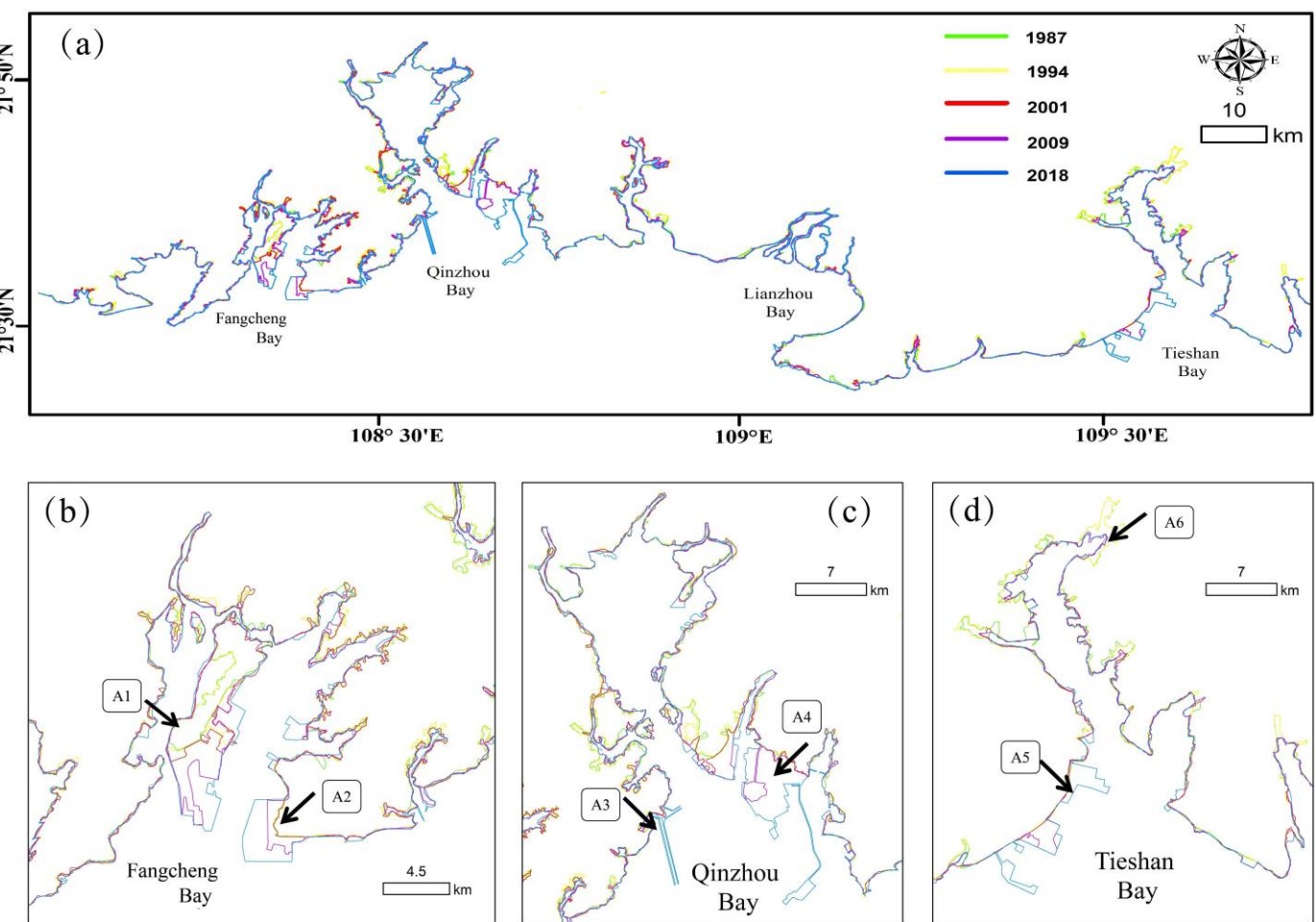

**Figure 6.** (**a**) The change in the MWBB in the Guangxi Beibu Gulf from 1987 to 2018. (**b–d**) The main areas were recorded by remote sensing observation: (A1) located in the port area of Fangcheng Bay, the MWBB continued to extend to the sea from 1987 to 2018. Remote sensing images show that the coastline development was used for urban development before 2001, and the port land was used after 2001. The reclamation area from 2001 to 2009, and 2009 to 2018, reached 12.7 km$^2$ and 5.8 km$^2$, respectively; (A2) located in an area of reclaimed industrial land in Fangcheng Bay, constructed after 2001, with a reclaimed area of 9.4 km$^2$; (A3) two long and narrow breakwaters extending more than 8 km to the sea, located in Qinzhou Bay; (A4) after 2009, a large reclamation project was undertaken in Qinzhou Bay with the aim of utilizing the area for urban and industrial purposes; (A5) three port areas built in Tieshan Bay, among which the middle one was built between 2001 and 2009, and the other two were built after 2009; (A6) a sprawling breeding pond situated in an estuary, covering an area of 6.9 km$^2$.

### 3.1.2. MWBB Type Change

As shown in Figure 7, the increase in aquaculture ponds, and industrial and urban areas, occupied large areas of coastal wetlands and coastline, which led to the total length of the MWBB in the Guangxi Beibu Gulf increasing from 1260.5 km (1987) to 1436.3 km (2018), while the natural boundary decreased by 281.4 km, and the artificial boundary increased by 457.2 km. In 1987, the MWBB was mainly natural landforms, such as bedrock, beaches, and estuaries, accounting for 87.8% of the total length of the Guangxi Beibu Gulf. Marine aquaculture was the main development activity, and the pond boundary accounted for 11% of its length. In 2001, the natural boundary of the Guangxi Beibu Gulf decreased

by 147.3 km, while the port boundary, urban and industrial boundary, protective dam boundary, and pond boundary, increased by 12.7 km, 28.1 km, 3.8 km, and 125.6 km, respectively. Among them, the pond boundary increased from 11% in 1987 to 21% in 2001. In 2018, the urbanization process in the coastal areas of the Guangxi Beibu Gulf continued to accelerate, with the natural boundary accounting for 58.2% of the total boundary length in 2018; the proportions of the urban and industrial boundary, port boundary, and protective dams along the coast of the Guangxi Beibu Gulf were increased from 3.6%, 1.1%, and 1.0% in 2001 to 6.0%, 5.8%, and 5.5% in 2018. In addition, the results show that the artificial boundary caused by the reclamation activities increased by 409.3 km, accounting for 89.5% of the new artificial coastline during 1987–2018.

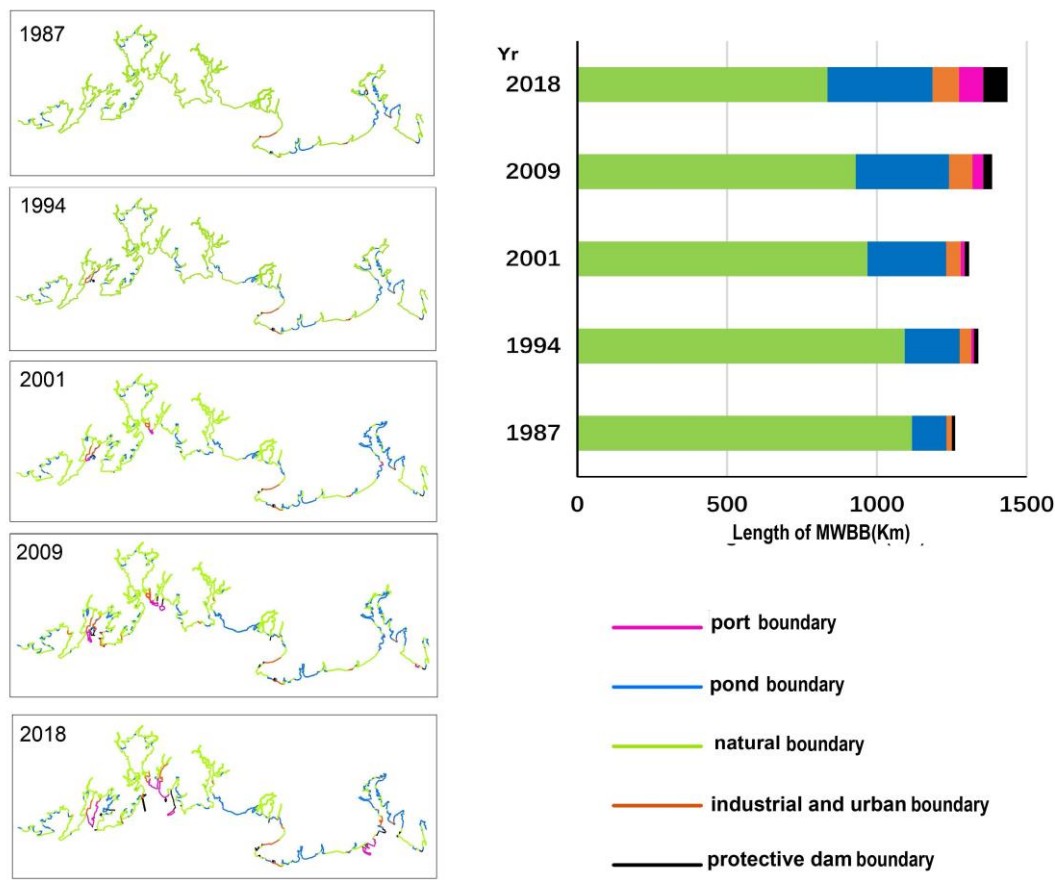

**Figure 7.** Spatio-temporal changes to the MWBB types in the Guangxi Beibu Gulf.

### 3.2. Spatio-Temporal Characteristics of Reclamation

Figure 8a illustrates the spatio-temporal distribution of the land use types in the reclamation areas between 1987 and 2018. Between 1987 and 2001, coastal reclamation was initially aimed at increasing fishery production. However, since the beginning of the 21st century, the focus of coastal reclamation has shifted towards promoting industrial development and urbanization. Hence, we divided the development process into two periods. The first period (1987–2001) saw a surge in large-scale reclamation activities, primarily for the aquaculture pond, with Tieshan Bay being the most prominent area. The second period (2001–2018) marked the pinnacle of the coastal development, with the construction of ports, wharves, and bridges along the coast, as well as the proliferation of large-scale industrial zones, resulting from the extensive land reclamation that spread throughout the Guangxi Beibu Gulf.

According to the statistics, the cumulative reclamation area increased by 163.8 km$^2$ between 1987 and 2018, with the main post-reclamation land use types being aquaculture ponds (45%), marine construction (23%), and industrial and urban areas (32%) (Figure 8b),

and the majority of reclamation taking place in Qinzhou Bay (42.5%), Tieshan Bay (21.3%), and Fangcheng Bay (20.7%). Over the study period, there was an exponential increase in Guangxi's coastal industries and port infrastructure, with the area occupied by marine construction and industrial and urban zones in the three cities expanding by almost 8-fold and 4-fold, respectively, at an annual average growth rate of 26% and 9% (Figure 8c).

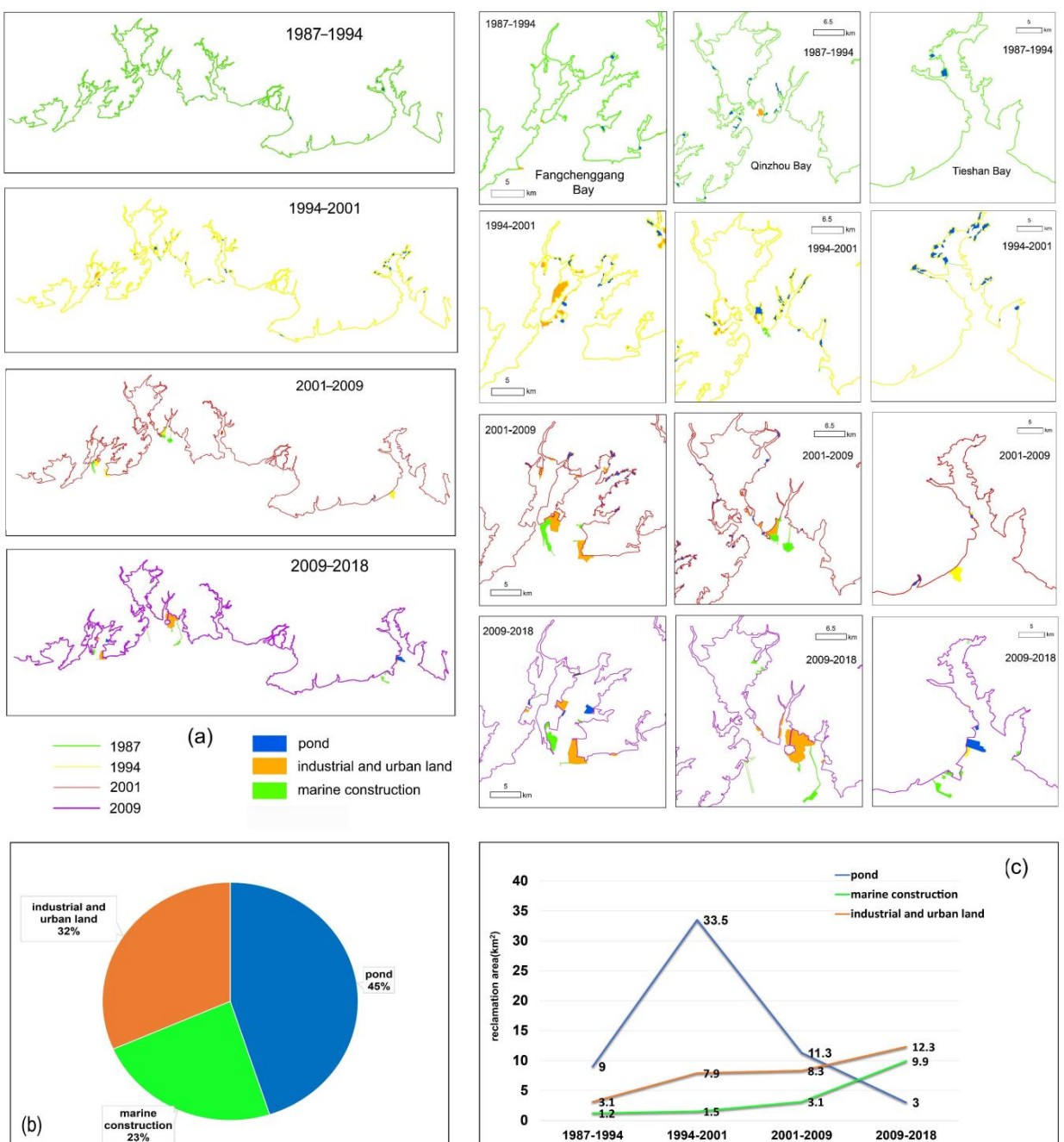

**Figure 8.** (**a**) The development process of reclamation in the Guangxi Beibu Gulf. (**b**) the area ratio of different reclamation types, and (**c**) the area grows in different stages.

### 3.3. Assessment on the Hydrodynamics

The effects on $M_2$: The co-tidal charts in 1987, 2001, and 2018 are shown in Figure 9a–c; the $M_2$ tide propagates mainly from the South China Sea into the Guangxi Beibu Gulf, and then a main branch of it propagates northwestwards into Tieshan Bay, while the small branch propagates westwards into Lianzhou Bay. When the tidal wave reaches Qinzhou

Bay, part of it turns into the Maowei Sea, and the other part spreads southwards to the area adjacent to the southwest of Qinzhou City. The largest amplitudes of $M_2$ tides can reach 70 cm in Tieshan Bay and the smallest $M_2$ amplitude (less than 6 m) appears in the Maowei Sea, caused by the characteristics of a wide interior and narrow mouth. These variations are consistent with the conclusions by Pan et al. [28]. The semi-diurnal tidal wave experienced considerable changes from 1987 to 2018 due to the coastline change (Figure 9d–f), and the change in the $M_2$ tidal amplitude was between –0.08 and 0.06 m. The maximum and minimum changes in the tidal amplitude appear in Sanniang Bay (located on the right side of Qinzhou Bay) and Tieshan Bay, respectively.

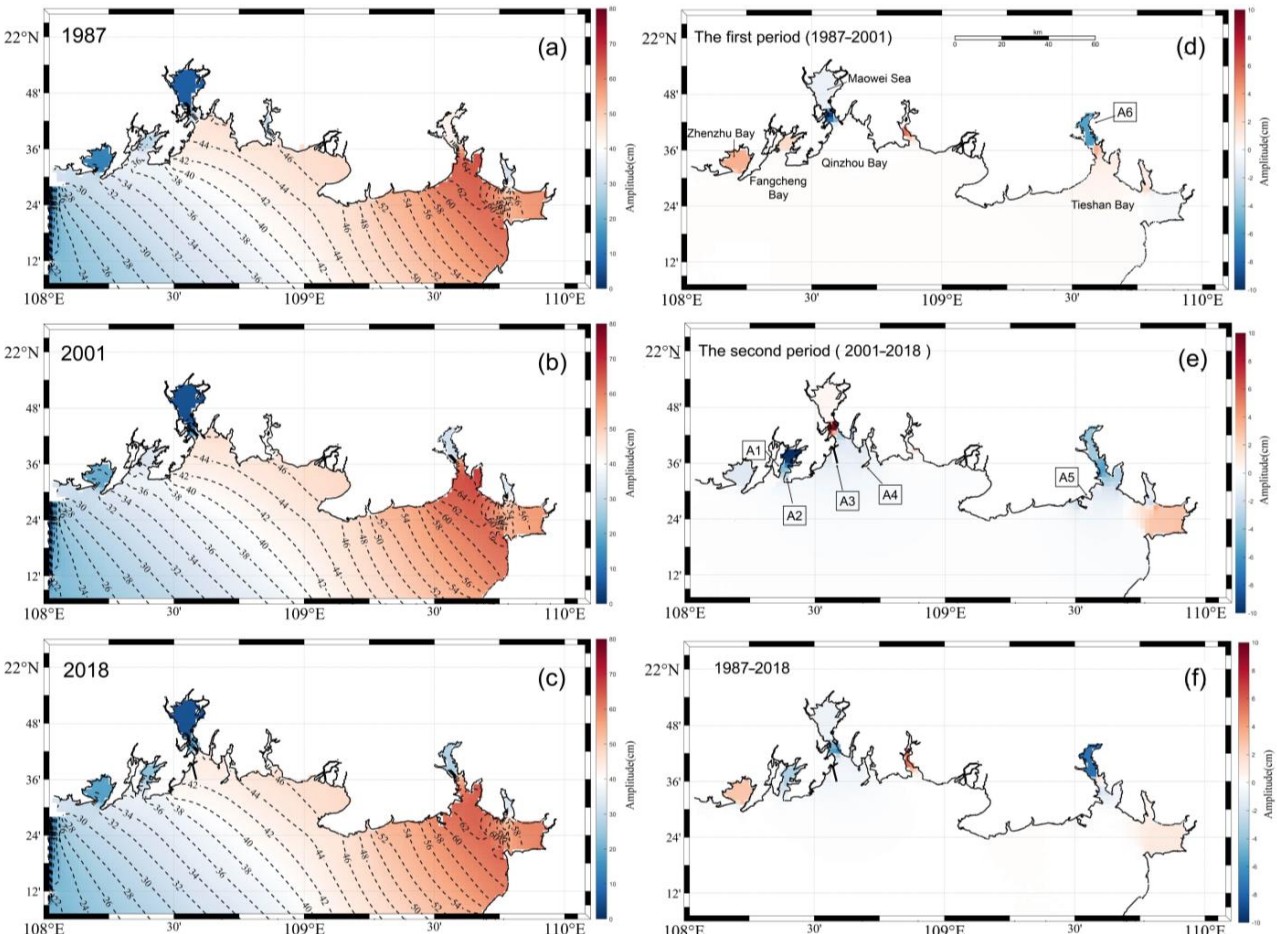

**Figure 9.** The co-tidal chart for $M_2$ is produced by the model, in 1987 (**a**), 2001 (**b**), and 2018 (**c**); the dotted black lines represent the phase lag (°). In the three years, the phase lag difference is small. Also, (**e**,**f**) compare the simulation results from 1987 and 2001 to those from 2018. (**d**) compare the 1987 simulation results to the 2011 results.

The effects on $O_1$ and $K_1$: The simulation results show that the amplitude of the diurnal tides ($O_1$ and $K_1$) in the Beibu Gulf are greater than that of the semi-diurnal tide ($M_2$). The $O_1$ and $K_1$ tides are diurnal tides, and their variation patterns are similar, although the $O_1$ tidal amplitude is greater than that of $K_1$ (Figures 10 and 11). The largest $O_1$ amplitude (greater than 80 cm) appears at the northeast mouth of Tieshan Bay. The amplitudes in the semi-enclosed bays of the Beibu Gulf (e.g., Maowei Sea and Zhenzhu Bay) are very small, less than 30 cm. The diurnal tidal wave experienced considerable changes from 1987 to 2018, due to the coastline changes. The $O_1$ tidal amplitude has changed by –0.14 to 0.11 m, and the change in the $K_1$ tidal amplitude was between –0.17 and 0.13 m. The maximum and minimum changes in the tidal amplitude appear in Zhenzhu Bay and the Maowei Sea, respectively.

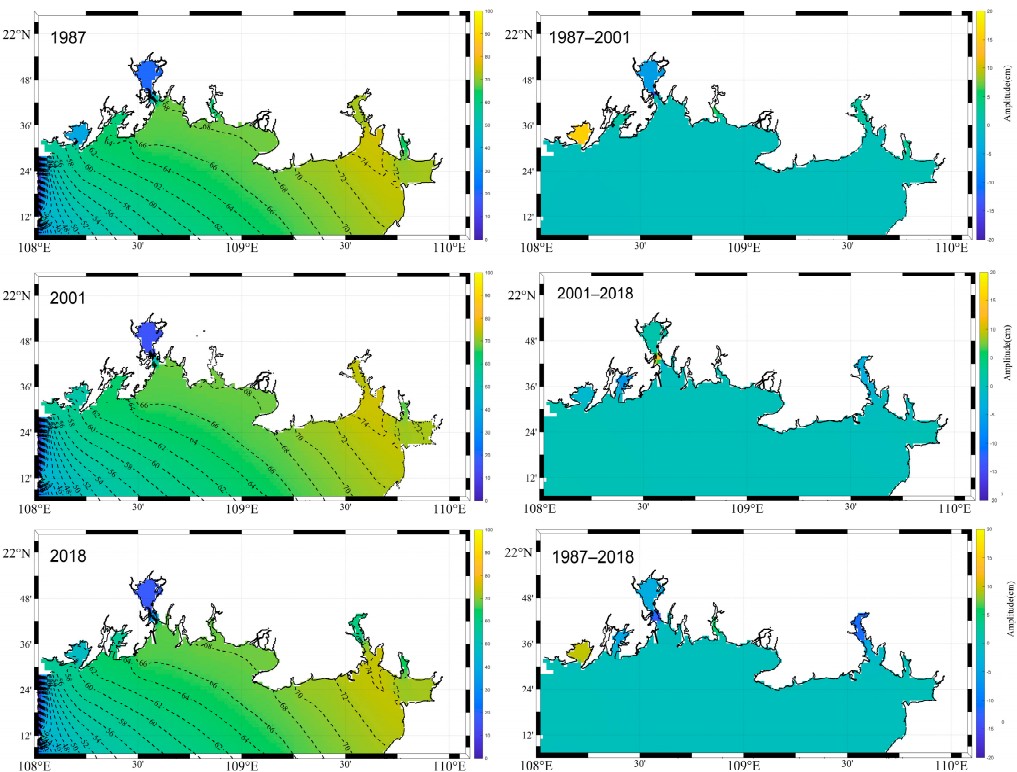

**Figure 10.** Same as Figure 9, but for O$_1$.

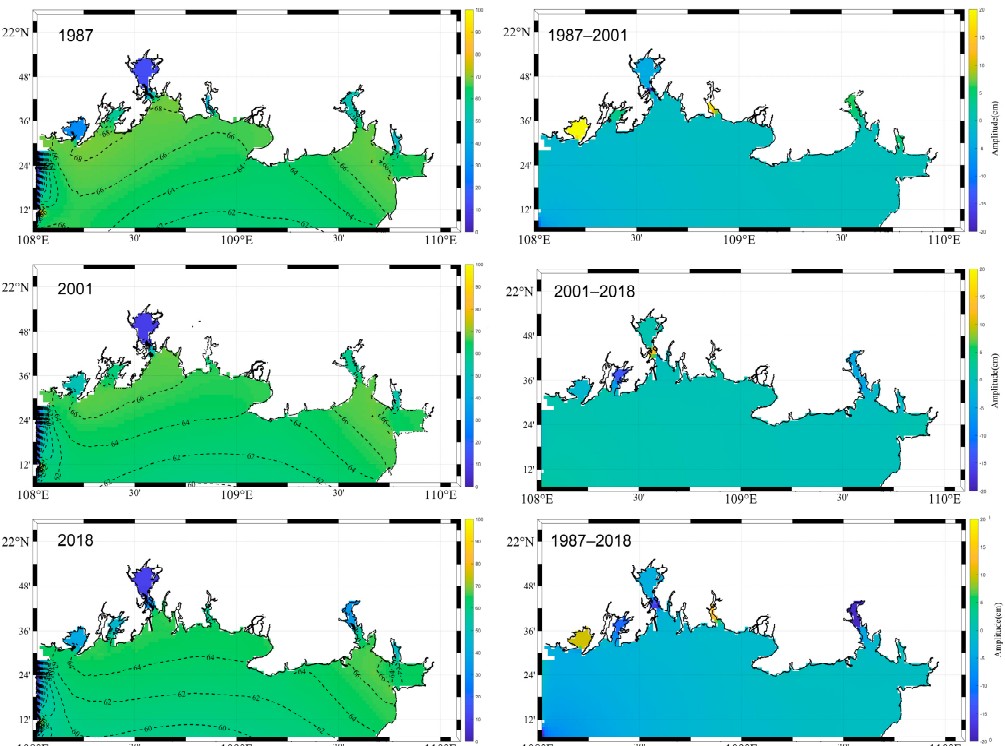

**Figure 11.** Same as Figure 9, but for K$_1$.

The effects on the residual current: In this paper, the Lagrange residual current principle is used to calculate the residual current. Figure 12 shows the distribution of the residual current in 2018. The strongest tidal current appears in Tieshan Bay, showing a clockwise rotation in the southern strait. Meanwhile, the second strongest tidal currents occur in the areas southeast of Zhenzhu Bay, exhibiting a counterclockwise direction. The

residual current experienced considerable changes from 1987 to 2018, due to the coastline changes. The magnitude of the flow velocity changed by 0.01 m/s to 0.03 m/s (Figure 13c). The maximum difference in the residual current appears in the channel between the Maowei Sea and Qinzhou Bay.

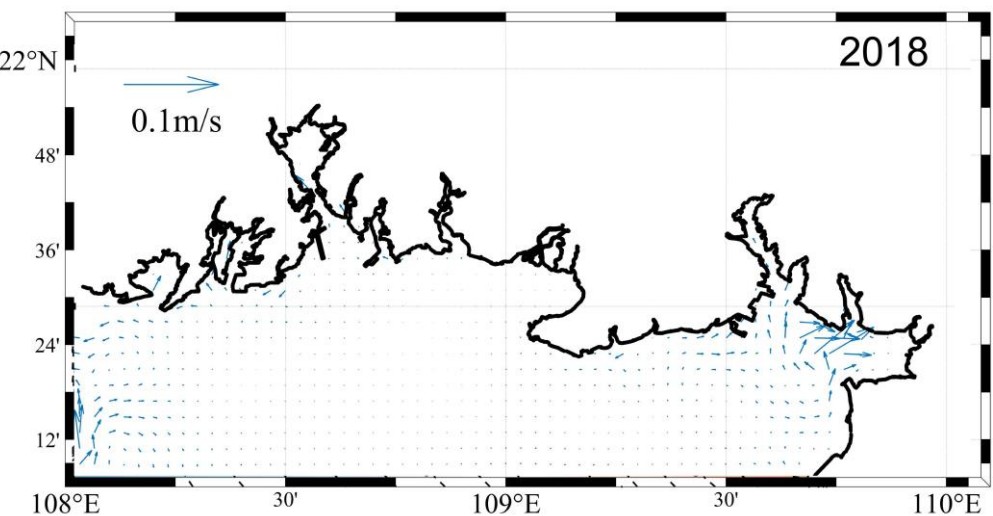

**Figure 12.** Model-produced tidal residual current in 2018.

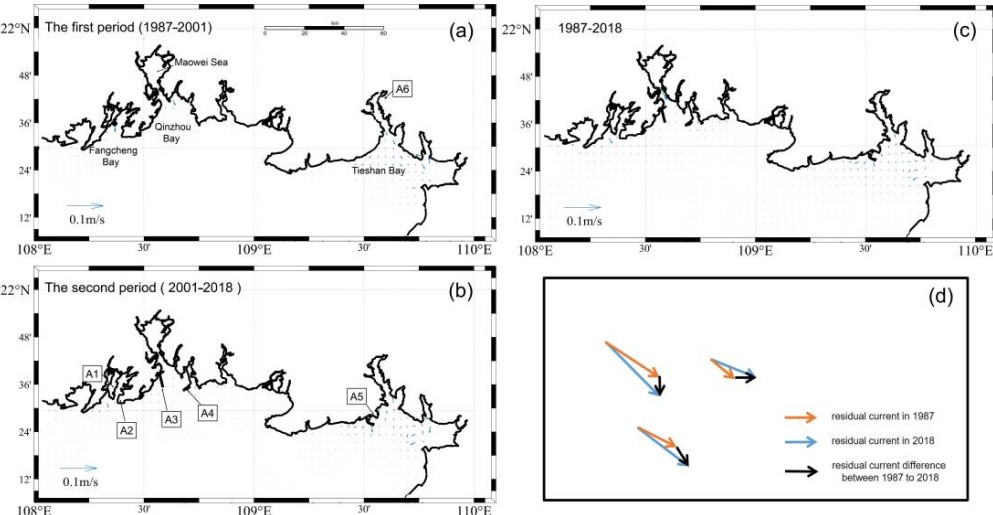

**Figure 13.** The difference in residual current for the three periods (**a–c**). Arrows denote the speed, and their lengths are determined by the magnitude of the residual velocity. The difference in residual current is based on flow vectors for two years (**d**).

In terms of different periods, the construction of the pond (A6) in Tieshan Bay had a significant impact on the $M_2$ and $O_1$ tidal component in the first period (1987–2001), reducing its amplitude by 8 cm and 5 cm within the bay, respectively (Figures 9d and 10). This variation led to an increase in seaward residual currents, with the maximum increase value exceeding 0.02 m/s. Furthermore, the construction caused a shift in flow directions downstream from the bay mouth and around its outlet (Figure 13a). This phenomenon was more obvious in the channel between Qinzhou Bay and the Maowei Sea. In the second period (2001–2018), the construction of the port area (A1) and the industrial land (A2) in Fangcheng Bay significantly decreased the amplitude of the $M_2$ and $O_1$ tidal component by 8 cm and 13 cm (Figures 9e and 10), producing a tidal residual current at the outer side with a velocity of around 0.01 m/s (Figure 13b). Additionally, the construction of two long, narrow breakwaters (A3) and a large-scale reclamation project for port use (A4) increased

the $M_2$ amplitude in the channel between the Maowei Sea and Qinzhou Bay by more than 8 cm. This is because the narrow reclamation projects limited the tide propagation upstream, causing the seawater to become congested in the channel. Lastly, the three ports (A5) on the westernmost side of Tieshan Bay reduced the amplitude of the $M_2$ and $O_1$ tidal components by 4 cm and 3 cm, respectively, leading to an overall velocity of less than 0.01 m/s and a counterclockwise tidal residual current (Figure 13b).

Overall, between 1987 and 2018, the reclamation in the Guangxi Beibu Gulf significantly decreased the amplitude of the $M_2$ and $O_1$ tidal constituent in the Tieshan Bay by 9 cm and 15 cm, the channel between the Maowei Sea and Qinzhou Bay decreased by 5 cm and 13 cm, and increased by 4 cm and 12 cm in Zhenzhu Bay (Figures 9f and 10). The variation in residual currents ranging from <0.01 m/s to 0.03 m/s took place in Tieshan Bay and Qinzhou Bay, with the predominant direction of the residual current being seaward, particularly in regions where narrow waterways have developed (Figure 13c). These findings align with the results by Chu et al. [12] and Shen et al. [43].

### 3.4. Driving Factors for Reclamation
#### 3.4.1. Sub-Indexes

Figure 14a shows that the artificial boundary and accumulated reclamation area significantly increased over 20 years. We further analyzed the trend in the sub-indexes increase from the statistical results on the socio-economic index in Guangxi Province between 1995–2015 and note a sharp increase after 2005, especially in the secondary industry GDP, the accumulated reclamation area, and the cargo throughput of major ports (Figure 14b,d). However, in contrast to the coastal GDP, the rate of population growth remained almost stable, while the urbanization degree shows slow but steady increases; from 1995 to 2015, the urbanization degree increased from 15.9% to 27.1%; both indexes did not match the rate of economic and reclamation area increase (Figure 14c).

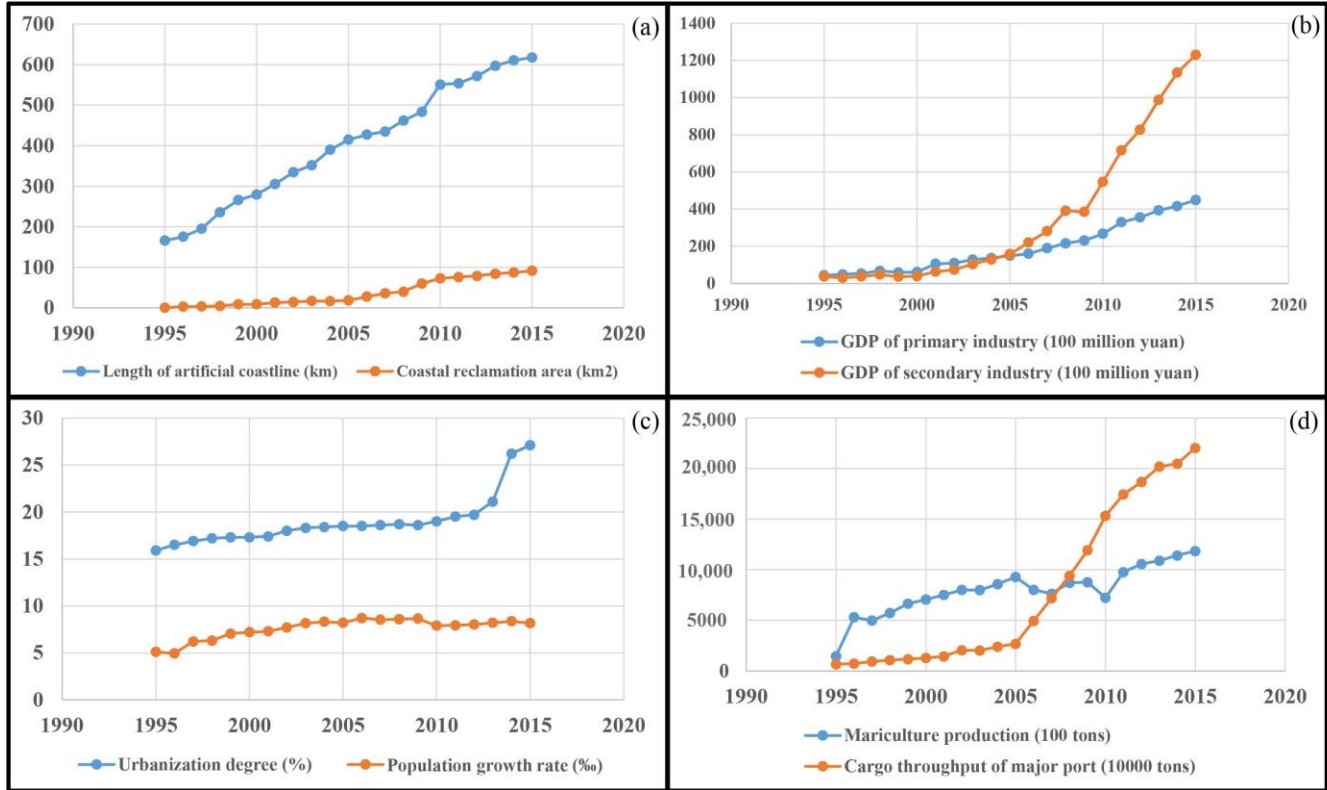

**Figure 14.** Growth trend for each indicator from 1995 to 2015. including length of artificial coastline and reclamation area(**a**), economic development (**b**), population growth (**c**), and marine industry development (**d**) in coastal cities.

3.4.2. Driving Force of Reclamation

In the SEM of the Guangxi Beibu Gulf reclamation models (Figure 15), the cumulative reclamation area is mostly influenced by economic development, including the positive effect of the GDP of primary industry (standardized path coefficient, $r = 0.408$), and the GDP of secondary industry ($r = 0.825$); the positive total effect of GDP indicated that the higher the GDP, the larger the reclamation area. Apart from economic development, the marine industry's growth is the most crucial contributing factor, with a positive effect ($r = 0.845$) via the cargo throughput of major ports, on the cumulative reclamation area. Population growth has a positive effect ($r = 0.364$) on the cumulative reclamation area via the urbanization degree. Moreover, the GDP from the secondary industry, the urbanization degree, mariculture production, and the cargo throughput of the major ports all showed positive effects on the length of the artificial coastline. Their path coefficients are 0.656, 0.364, 0.691, and 0.711, respectively. The cumulative reclamation area has positive effects ($r = 0.993$) on the length of the artificial coastline, indicating an increase in the reclamation area with an increase in the artificial coastline.

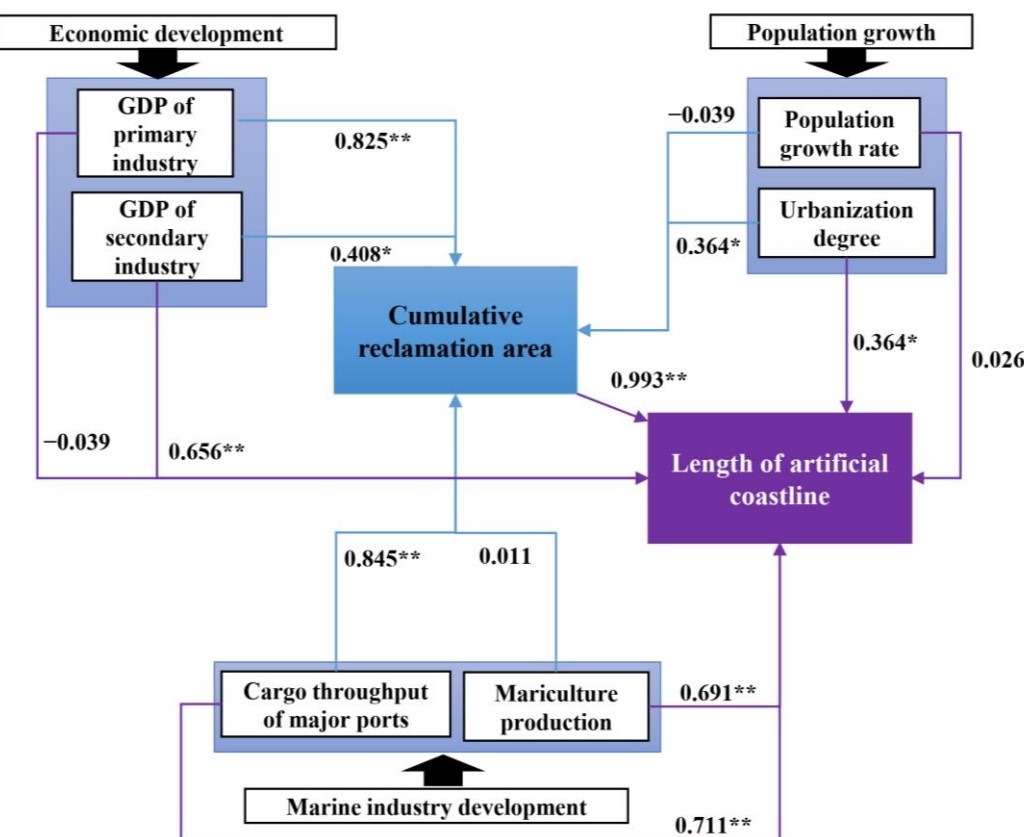

**Figure 15.** SEM to estimate the magnitude and significance of relationships between population growth, economic development, marine industry development, and reclamation. * $p < 0.05$, ** $p < 0.01$.

## 4. Discussion

### 4.1. Evaluation of MWBB Change

With high-intensity human activities, such as reclaiming land from the sea, the share of natural shorelines has already decreased drastically from 87.8% in 1985 to 58.2% in 2018. Our findings align more closely with the detailed regional analysis of remote sensing data collected at various times over 37 years by Sun et al. [44], who found that the natural coastline retention ratio reduced to 57.97% in 2017. In addition, the reclamation assessment results for the study area are compared with the spatial distribution of the reclamation and coastline published by Yu et al. [45] over the same period. The results show the cumulative area also reached 9686.02 hm$^2$ in 2018, and the spatial distribution of the reclamation is

mainly concentrated in Tieshan Bay and Qinzhou Bay. However, this result is different from ours on the Guangxi Beibu Gulf, this discrepancy could be explained by the more explicit coastline corrections applied in our method, which further considered the coastline changes caused by aquaculture ponds in Tieshan Bay that took place mainly in the 1990s.

### 4.2. Evaluation of Hydrodynamics Change

To the best of our knowledge, there have not been enough studies conducted on how coastal changes affect the tidal dynamics in the Beibu Gulf. These geographical and temporal tidal biases may be unintentionally exposed in regional-scale earth observation studies that compare tidal dynamics without accounting for coastline change, potentially leading to erroneous estimates on tidal change [46]. Due to the computationally intensive nature of the fine mesh in the numerical model, earlier work has mostly been limited to local applications in a portion of the Beibu Gulf. Jiang et al. [47] indicated that the change in the shoreline made the flow velocity of the water near the top of the Sandun Highway increase by 0.2 m/s. Wang et al. [25] further demonstrated that the change in the coastline caused by reclamation resulted in a decrease of 0.12 m/s in the flow velocity in the east Maowei Sea. Our results further complement the understanding on the hydrodynamics of Beibu Gulf reclamation. Our model shows the $M_2$ tidal amplitude has been decreased and the seaward residual current has been increased due to reclamation, especially in some bays where marine construction has been formed, such as in Tieshan Bay and Qinzhou Bay. We are able to properly identify medium-resolution square meshes (spatial resolution is $1/60° \times 1/60°$) where the shoreline migration mechanism does not adequately adjust the tide impacts, by integrating remote sensing technology and hydrodynamic simulation technology. The efficiency of this method is shown in Figures 10–13, where a coastline alteration in the Beibu Gulf affects the tide in the coastal waters.

### 4.3. Evaluation of the Conceptual Framework

This paper developed a process-based "Social Development & Human activity & Tidal environment" framework to track the evolution of the tide for coastline change and to simulate the response of artificial coastlines to their resultant socio-economic situation in the Guangxi Beibu Gulf, over the last 30 years. Above all, we confirmed that the relationship between the three aspects is equivalent to three sets of gears that fit together, with socioeconomic factors as the starting point, propelling the increase in human activities (i.e., land reclamation), resulting in changes in the tidal environment (see Figure 3).

#### 4.3.1. Social Development and Human Activity

The SEM model showed that the secondary industry GDP ($r = 0.825$) and cargo throughput of major ports ($r = 0.845$) have positive effects on the cumulative reclamation area. According to the Marine Statistical Yearbook of China, the cargo throughput of major ports in the Beibu Gulf has increased 23.5-fold. Meanwhile, the scale of reclamation is continuously being expanded. Compared with that in 1987–2001, the area in 2001–2018 increased 2.4-fold. On the other hand, the cargo throughput of major ports ($r = 0.711$) and mariculture production ($r = 0.691$) have positive effects on the length of the artificial coastline. The area of marine aquaculture in the coastal waters of Guangxi has increased every year, from 2001 to 2018, with the area of aquaculture increasing by 1.4-fold, compared with that from 1987 to 2001, resulting in a substantial increase in the production of aquatic products. From 2001 to 2018, mariculture production increased 8.1-fold. The development of the port, fishery products, and the associated transportation industry, promote rapid expansion of the reclamation area.

In the SEM model, the cumulative reclamation area has positive effects ($r = 0.993$) on the length of the artificial coastline, indicating that the reclamation area has continually increased as a direct response to economic and industrial growth, as well as the steady increase in length of the artificial shoreline. This happened for several reasons, including government policy amendments, and road and airport construction; for example, Sandun

Road is being constructed to the east of the Qinzhou Bonded Port, in order to stimulate natural siltation for future land reclamation and port development.

### 4.3.2. Human Activity and Tidal Environment

Industrial construction, oil exploitation, dikes, and bridge building, among other human reclamation operations, alter not just the coastline, but also the depth of the water, which may significantly alter tidal evolutions in this estuarine–coastal system [40]. The results from this study reveal that the continuous land reclamation over the past 30 years has altered the tidal amplitude and has greatly changed the residual current variation in the Guangxi Beibu Gulf. The residual current deflects to the seawall side due to the reclamation's coverage of shallow areas, narrowing the cross-sections, which is consistent with previous studies [48]. Additionally, rapid coastline change has resulted in the $M_2$ and $O_1$ tidal amplitudes having changed by up to 0.08 m and 0.15 m, thereby increasing coastal risks, such as storm surges. Meanwhile, places with increased fast-rising tides, as a result of reclamation projects, have an impact on ports, harbors, and estuaries, resulting in severe siltation.

### 4.4. Implications for Future Coastal Management

During our study period, a large number of synthetic engineering structures were used in the Guangxi Beibu Gulf. The Fangchenggang Nuclear Plant, which was built on the western edge of the outer harbor, is an illustration. Despite having a surface area of only 1.2 km$^2$, two impervious 18 km long dikes that are used to emit thermal effluent into the bay prevent the exchange of water between the two sides [49]. The natural ecosystems of the ocean and land, including grasses, trees, and biological communities, are destroyed during the construction of these maritime structures, and it is difficult to evaluate the potential loss of ecological advantages. Engineering solutions based on marine ecology may be applied to lessen this conflict. For instance, replanting and restoring coral reefs, shellfish, or salt marshes [50]. Further, we propose that reclamation should be stopped completely, especially in Fangcheng Bay and Qinzhou Bay, which have experienced relatively serious reclamation impacts, in order to use the sea space according to the overall scientific plan, strictly control the increase in the reclamation area every year, and pay attention to the protection and restoration of coastal mangroves, seagrass beds, coral reefs, and other special habitats.

### 5. Conclusions

In this paper, we reveal the spatio-temporal evolution of the MWBB in the Guangxi Beibu Gulf based on a 5-year record of remote sensing images in the most recent 40 years. To simulate the tidal movement, we incorporated TOPEX/Poseidon (T/P) satellite altimeter data into the adjacent model and provided the bottom friction coefficient and ideal coastline position extracted from remote sensing technology. We found that reclamation projects have had significant impacts on the variation in tidal amplitude, resulting in increased seaward residual currents in the Guangxi Beibu Gulf. The main findings from our study are:

(1) The Guangxi Beibu Gulf's MWBB has changed dramatically between 1987 and 2018, owing to the construction of ports and ponds. Reclamation initiatives have damaged approximately 20–25% of the shore's water area, most notably in Qinzhou Bay. A large reclamation project is being developed out into the bay at Qinzhou City, and this trend is influencing the land use pattern in the coastal zones.

(2) The total reclamation area increased by 163.8 km$^2$ between 1987 and 2018. Aquaculture ponds (45%), marine construction (23%), and industrial and urban areas (32%), are the primary post-reclamation land use types. The SEM-based reclamation effect model shows that the spatio-temporal changes caused by the reclamation are the result of the comprehensive influence of the population growth, economic development, and marine industries in Guangxi coastal cities.

(3) Reclamation projects have had a great influence on the amplitude of the tidal constituent near the offshore sea, especially in some bays where marine construction has formed, such as in Tieshan Bay and Qinzhou Bay. The change in the coastline made the amplitude of the $M_2$ tidal constituent in Tieshan Bay decrease by 6 cm to 8 cm, the channel between the Maowei Sea and Qinzhou Bay decrease by 4–8 cm, and the variation in residual currents from <0.01 m/s to 0.03 m/s occurs in the Tieshan Bay and Qinzhou Bay, with higher values observed near the outlets.

In order to offer suggestions for solutions to the identified coastal changes in the study area, we also mention coastal zone management techniques for use in industrial development plans and protection ideas. The findings from this study could offer some guidelines for the protection and development of the changing coasts that are subject to significant human intervention.

**Supplementary Materials:** The following supporting information can be downloaded at: https://www.mdpi.com/article/10.3390/rs15215210/s1, Table S1: Evaluation index system of the driving force factors of the reclamation; Table S2: Related index data of Guangxi Beibu Gulf.

**Author Contributions:** J.L.: Formal analysis, Methodology, Software, Writing—original draft, Writing—review & editing. Y.Z.: Data curation, Formal analysis, Visualization. R.C.: Software, Validation. X.L.: Methodology, Resources; M.X.: Conceptualization, Funding acquisition, Investigation, Resources. G.G.: Supervision, Resources; Q.L.: Validation, Writing—review & editing. All authors have read and agreed to the published version of the manuscript.

**Funding:** This work was supported by National Key R&D Program of China (2021YFB3901300) and the National Natural Science Foundation (NSFC, No. 42006004). Thanks go to China-Indonesia Maritime Cooperation Fund "Construction of Ecological Marine Ranching in Indonesia" and the Sino-Indonesia Joint Laboratory for Marine Sciences (SIMS).

**Data Availability Statement:** The bathymetry data come from the General Bathymetric Chart of the Oceans (https://www.gebco.net/data_and_products/gridded_bathymetry_data/, accessed on 23 June 2023). The data of flow fields are available at http://apdrc.soest.hawaii.edu/las/v6/dataset?catitem=2985 (accessed on 23 June 2023).

**Conflicts of Interest:** The authors declare no conflict of interest.

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
