# Peer review of "Enhanced Impact of Land Reclamation on the Tide in the Guangxi Beibu Gulf"

_remotesensing, doi:10.3390/rs15215210_

Round 1
Reviewer 1 Report
Comments and Suggestions for Authors
Authors aim to address the following issues
1) to draw the spatio-temporal changes of 86 MWBB and reclamation in the Guangxi Beibu Gulf; 2) to explore if regime shifts in tidal 87 dynamics can occur in a semi-enclosed gulf such as Beibu Gulf under reclamation impact; 3) to evolution of the potential effects of reclamation projects; 4) to explain the driven 89 mechanism of the reclamation.
Points 1 and 2 are clearly presented in the text. Although points 3 and 4 are not. More discussion and a better presentation of the SEM model are needed to address this.
Authors need to describe the SEM method with more details and link the result in the discussion
Reviewer 2 Report
Comments and Suggestions for Authors
This manuscript mainly provides a shoreline analysis over past 40 years by so-called MWBB methods. The numerical modelling methods have been applied to study the change of tidal wave propagation under different shoreline conditions. Also, several analyses have been done by relating the economic development with the shoreline change. l have some questions that need clarification.
1. Line 130: Resolution of the satellite image is 30m. What is the annually averaged rate of the coastline progradation or retreat, and are there spatial differences in the study area? Does the resolution of satellite images affect the identification of coastlines?
2. Regarding to the MWBB method, to my knowledge, the tidal waver level is required to identify the water line when the satellite passes over. But by reading the present manuscript, I’ve no idea how the authors do this. And what is the tool “streamline”? The illustration of the MWBB method is poor here reducing the readability of the article.
3. Line 188: about the tidal model. Even though the authors mentioned that, the simulation method has been proposed in a previous published paper, I suggest at least the basic information of the model should be illustrated here. Include the position of the observation point, open boundary conditions and so on. in the present manuscript
4. Figure 5. In the model with different year, I assume that the authors only change the shoreline but keep the bathymetry as unchanged. So, why comparing the tidal data (harmonic constants) with different year for model verification? Furthermore, to my knowledge, the diurnal tide in the study area should be also important. What is the model performance for diurnal tide?
5. Line 333: The author only analyzed the changes in M2 tide when analyzing the data. Again, how about the diurnal tide? How does the authors calculate the residual current?
6. Figure 11: How is this Figure drawn? Generally speaking, when comparing the variations in residual flow field, we can compare the differences in the magnitude of flow velocity or the difference in flow vectors. The result presentation in Fig 11 is hard to understand. Please clarify.
7. Regarding the SEM method, it is also required to illustrate the method in detail.
Round 2
Reviewer 1 Report
Comments and Suggestions for Authors
Accept after the new changes
Reviewer 2 Report
Comments and Suggestions for Authors
The authors have addressed all my comments and the revised manuscript has been improved significantly.